# Mosquitoes (Diptera: Culicidae) in the Dark—Highlighting the Importance of Genetically Identifying Mosquito Populations in Subterranean Environments of Central Europe

**DOI:** 10.3390/pathogens10091090

**Published:** 2021-08-26

**Authors:** Carina Zittra, Simon Vitecek, Joana Teixeira, Dieter Weber, Bernadette Schindelegger, Francis Schaffner, Alexander M. Weigand

**Affiliations:** 1Unit Limnology, Department of Functional and Evolutionary Ecology, University of Vienna, 1090 Vienna, Austria; carina.zittra@univie.ac.at; 2WasserCluster Lunz—Biologische Station, 3293 Lunz am See, Austria; simon.vitecek@wcl.ac.at (S.V.); bernadette.schindelegger@wcl.ac.at (B.S.); 3Institute of Hydrobiology and Aquatic Ecosystem Management, University of Natural Resources and Life Sciences, Vienna, Gregor-Mendel-Strasse 33, 1180 Vienna, Austria; 4Zoology Department, Musée National d’Histoire Naturelle de Luxembourg (MNHNL), 2160 Luxembourg, Luxembourg; joanamtl@gmail.com (J.T.); dieter.weber124@gmx.de (D.W.); 5Francis Schaffner Consultancy, 4125 Riehen, Switzerland; francis.schaffner@uzh.ch

**Keywords:** *Culex* *pipiens* s. l., *Culex* *torrentium*, *Culiseta glaphyroptera*, caves, subterranean environment, Luxembourg, Germany

## Abstract

The common house mosquito, *Culex* *pipiens* s. l. is part of the morphologically hardly or non-distinguishable *Culex* *pipiens* complex. Upcoming molecular methods allowed us to identify members of mosquito populations that are characterized by differences in behavior, physiology, host and habitat preferences and thereof resulting in varying pathogen load and vector potential to deal with. In the last years, urban and surrounding periurban areas were of special interest due to the higher transmission risk of pathogens of medical and veterinary importance. Recently, surveys of underground habitats were performed to fully evaluate the spatial distribution of rare members of the *Cx.* *pipiens* complex in Europe. Subterranean environments and their contribution to mosquito-borne pathogen transmission are virtually unknown. Herein, we review the underground community structures of this species complex in Europe, add new data to Germany and provide the first reports of the *Cx. pipiens* complex and usually rarely found mosquito taxa in underground areas of Luxembourg. Furthermore, we report the first finding of *Culiseta glaphyroptera* in Luxembourg. Our results highlight the need for molecular specimen identifications to correctly and most comprehensively characterize subterranean mosquito community structures.

## 1. Introduction

The globally distributed *Culex pipiens* complex (or *Culex pipiens* assemblage *sensu* Harbach, 2012) consists of several taxa: *Cx. quinquefasciatus*, *Cx. pipiens pallens*, *Cx*. *australicus* and the nominate taxon *Culex pipiens* in Europe [1]. The latter taxon includes two behaviorally and genetically distinct forms (‘f.’), *Cx. pipiens* f. *pipiens* and *Cx. pipiens* f. *molestus*, that do not differ morphologically and are able to hybridize in areas of coexistence [2]. Members of the complex are of medical importance as they are primary vectors of several pathogens, including the West Nile virus, the widespread cause of arboviral neurological disease, and are often the most abundant mosquitoes in urbanized areas [3]. Emergence, distribution and transmission of the West Nile virus and other mosquito-borne pathogens are regulated through potential vector communities that link suitable reservoirs and susceptible hosts [4,5]. Differences in vectorial capacity and vector competence between the *Cx. pipiens* complex members due to specific ecology, physiology, and behavior, therefore, have direct reverberations on animal and human health [6].

Forms of *Cx. pipiens* are reported to differ, in which eurygamous *Cx. pipiens* f. *pipiens* requires more space for mating than stenogamous *Cx. pipiens* f. *molestus*, which does not mate in swarms. These observations led to the general assumption that *Cx. pipiens* f. *pipiens* is restricted to epigean (above-ground) sites, while *Cx. pipiens* f. *molestus* is considered to inhabit mostly hypogean (underground, subterranean) sites, especially such close to human settlements [7]. Further, different host and habitat preferences of both forms and their hybrid offspring potentially lead to distinct roles in host-vector-pathogen dynamics: In contrast to *Cx. pipiens* f. *molestus*, *Cx. pipiens* f. *pipiens* is reported to prefer avian hosts, while the host and habitat preference of their cross-bred offspring is not sufficiently known. Additionally, they have contrasting strategies to survive winter, where female *Cx. pipiens* f. *molestus* remain active but female *Cx. pipiens* f. *pipiens* overwinter undergoing diapause in shelters associated with human settlements like cellars or attics [8].

Different populations of the *Cx. pipiens* complex were discovered earlier to inhabit fully enclosed sites, but also crevices connected to above-ground habitats or open-air habitats [9]). However, the allozyme loci used then to differentiate between taxa precluded identification of *Cx. pipiens* forms [10,11]. Once standardized and replicable molecular methods [12,13] allowed reliable differentiation of the *Cx. pipiens* complex members, this taxon was examined in Europe in epigean sites [2,14]. In contrast, the composition and seasonality of the *Cx. pipiens* complex in relation to abiotic parameters of subterranean resting and hibernation habitats were often neglected.

Subterranean sites such as natural caves, mining galleries, tunnels, and culverts are resting and hibernation shelters for several subtroglophilous insects of the order Diptera, not forming permanent subterranean populations but seasonally inhabiting underground habitats [15,16,17]. Mosquitoes of the genus *Culex* are known to use non-urban subterranean habitats as hibernation and resting sites. The purported adaptation to urban environments, therefore, has been discussed for decades [9,17]. Amongst the earliest reports is that of Legendre [9], who observed the larval development of *Cx. pipiens* s. l. in well-connected underground cave systems with strong exchange with above-ground habitats if water and air temperatures were suitable and nutrients available [9]. Subterranean sites provide stable and adequate conditions (in terms of humidity and temperature) and are visited actively by a range of mosquito taxa, probably to avoid unfavorable conditions such as dryness and cold temperatures or to reduce predation pressure [17,18]. This possibly is a behavioral adaptation but seems to be a common trait in mosquitoes, and while above-ground distribution patterns of the *Cx. pipiens* forms, their offspring and *Cx. torrentium* were recently examined in Germany and Austria [2,14,19], the hypogean distribution of these taxa remains obscure at greater scales. Life in caves in Central Europe is well-known, particularly in Germany and Luxembourg [15,16,20,21,22,23,24,25,26,27], but mosquitoes collected or spotted in caves were often mostly identified to family-level ([16,28], long-term collection data for [29]). With regard to members of the *Cx. pipiens* complex—a taxon reported as widely distributed and highly abundant in underground habitats [30]—the general practice of reporting combined occurrence data for *Cx. pipiens* s. l. and *Cx. torrentium* as “Culicidae” or even “*Culex pipiens*” is not ideal given their potentially different distribution patterns and epidemiological relevance [31].

Within the *Cx. pipiens* complex, *Cx. pipiens* f. *molestus* is generally accepted as the hypogean counterpart of *Cx. pipiens* f. *pipiens* [10,32], but both forms are known to occur in sympatry above-ground [2]. Yet, it appears that they can become strongly isolated, as observed in the London underground railway system where a dominance of f. *molestus* was found [10]. Generalizations and expected distribution patterns extrapolated from these records were, however, not confirmed as members of the *Cx. pipiens* complex were found in sympatry in subterranean habitats of both urban and rural areas in Austria, Germany, and Hungary [18,29,33]

At present, mosquito community composition, including members of the *Cx. pipiens* complex, in the subterranean realm, has been rarely studied using molecular tools, and baseline data [18,29,33] are slowly emerging in Europe. Better knowledge about the distribution and composition of mosquitoes and the *Cx. pipiens* complex in particular, is crucial to assess vector-borne pathogen dynamics in rural habitats and to estimate the potential impact of to date neglected subterranean sites on public health. In this contribution, we review available literature and present new data to provide the first synopsis on mosquitoes of the *Cx. pipiens* complex in underground environments and provide a summary of Culicidae in subterranean habitats.

## 2. Results

In Germany, 151 specimens belonging to the *Cx. pipiens* complex were molecularly analyzed (Table 1). A total of 56% were found in the transition zone, and most of the specimens were collected in autumn (Appendix A). *Culex pipiens* f. *pipiens* was most abundant with 99 specimens including two males, found exclusively in the transition zone. *Culex pipiens* f. *molestus* was rarely represented by two specimens found in autumn in the Westerwald and the Swabian Jura again in the transition zone. Hybrids were represented by four specimens collected in spring and autumn at the Swabian Jura. A total of 46 specimens were identified as *Cx*. *torrentium*.

In Luxembourg, 159 mosquitoes belonging to the *Cx. pipiens* complex were found in subterranean areas in Luxembourg. *Culex pipiens* f. *pipiens* was the most abundant taxon with 119 specimens (75%), followed by *Cx*. *torrentium* (38 specimens; 24%) and *Cx. pipiens* f. *molestus* (2 specimens; 1%). Hybrids of *Cx. pipiens* f. *pipiens* and *Cx. pipiens* f. *molestus* were not detected in Luxembourg samples.

We first collected *Culiseta annulata* and *Cs. glaphyroptera* in Luxembourg caves, representing at the same time the first record of *Cs. glaphyroptera* in Luxembourg (Table 2). Studies of subterranean mosquito populations in Europe are available from Austria [30,33,34], Germany [23,24,25,26,27,28,29,33], Croatia [35], Czech Republic [34,36,37], France [38], Hungary [18,34], Italy [39], Luxembourg [16], Poland [40], Norway [17], Slovakia [41,42,43], and Sweden [44] (Table 2). Several mosquito taxa are reported from subterranean habitats at larger geographical scales (i.e., spanning more than three European countries), including *Anopheles maculipennis* s. l., *Cs. alaskaensis*, *Cs. annulata*, *Cs*. *glaphyroptera*, and members of the *Cx. pipiens* complex. The highest *Culex* diversity in subterranean habitats was found in Austria and Germany, comprising seven and five taxa, respectively. Additionally, unregular occurrences of other mosquito taxa in subterranean habitats are reported: *Aedes cinereus*/*geminus*, *Ae. cataphylla*, *Ae*. *rossicus*, *An. messae*, *An. claviger*, *Cx. hortensis*, *Cx. modestus*, *Cx. territans*, and *Uranotaenia unguiculata* (but not *An. marteri* that was previously reported from subterranean sites in Hesse, Germany [29], due to a database error and should be considered reports of *An. maculipennis* s. l.).

## 3. Discussion

It appears that molecular analysis of hardly or un-identifiable *Culex* species collected in natural and artificial caves is not common despite the epidemiological importance of these taxa. Countries where morphological identification and the use of molecular tools were combined (Austria, Germany) recovered higher diversity, but data on numbers of molecularly identified specimens are provided here (Appendix A) and the Austrian study [33] only. Resolving the *Cx. pipiens* complex with molecular methods, we found no indication for a greater proportion of *Cx. pipiens* f. *molestus* in underground habitats. This is in line with previous reports on these taxa, that appear to be present at relatively constant frequencies in above- and underground habitats as well as over long time periods in Central Europe [2,29,33]. We found *Cx. pipiens* f. *pipiens* to be the dominating mosquito in the investigated natural and artificial subterranean sites (Appendix A). Thus, data at hand rather indicate a sympatric occurrence of the two forms in above- and underground sites. While it may be possible that in cities with significant underground constructions like London or Helsinki, such isolation may take place, we found no evidence for reproductively isolated *Cx. pipiens* f. *pipiens* and *Cx. pipiens* f. *molestus* populations in Central European artificial and natural subterranean shelters. Intriguingly, the proportions of *Cx. pipiens* f. *pipiens*, *Cx. pipiens* f. *molestus* and *Cx. pipiens* f. *pipiens* X f. *molestus* hybrids in both above-ground and subterranean habitats appear to mirror patterns that would be expected under Hardy–Weinberg equilibria of two alleles in a panmictic population [19,45]. Under reproductive isolation, patterns strongly deviating from such an equilibrium would have been expected in underground habitats. The presumed reproductive isolation of the *Cx. pipiens* forms by niche differentiation, therefore, cannot be corroborated. However, to effectively test for reproductive isolation in underground habitats by assessing deviations from Hardy–Weinberg equilibrium-like states, far greater numbers of specimens need to be analyzed. More frequent and more specific sampling in underground habitats, including the collection of potentially present mosquito larvae, should be conducted to assess potential reproductive isolation between *Cx. pipiens* forms.

The subterranean realm seems to harbor a specific mosquito community recruited from the surrounding above-ground habitats. However, the available data are too limited to speculate about colonization pathways or how subterranean habitats are used. Data at hand indicate that several *Culex* and *Culiseta* species regularly use subterranean sites, sometimes accompanied by *Anopheles* species, *Uranotaenia unguiculata*, and exceptionally by *Aedes geniculatus*, a tree-hole breeding species (collected once, in summertime) (data presented here [20,34,46]). At the same time, the new records of *Cs. annulata* and *Cs*. *glaphyroptera* from Luxembourg demonstrates that cave habitats can be important sites for mosquito monitoring, especially when combined with molecular tools. Amongst the taxa using caves, *Cx. pipiens* f. *pipiens* reaches highest abundances [29,33]. Another *Culex* species, *Cx. torrentium*, occurs regularly and in quite high abundances in underground habitats, despite its apparent rarity in epigean habitats. In this study, we could confirm this pattern in Luxemburg and Germany in congruence with previous findings in Germany [29] and Austria [33]. However, restricted access to molecular tools seems to impede assessments of mosquito communities in caves: While there is ample information about the occasional or regular occurrence of a wide range of taxa in subterranean habitats, there is little data on the *Cx. pipiens* complex or morphologically similar taxa. Apparent absences of *Cx. torrentium* from Hungarian caves in the Bakony Balaton region, parts of Germany, or the Czech Republic could result from such limitations [18,28,36].

In addition to the lack of taxonomic resolution in the available data, the cave habitats and their potentially relevant characteristics (entrance size, temperature regimes, humidity, presence and permanence of aquatic habitats, etc.) are poorly described. Such data are necessary to evaluate which parameters drive hypogean mosquito community composition and abundance patterns. Land cover was previously shown to control mosquito community assembly at larger scales [5], but which processes lead to cave-use in mosquitoes is not clear. Data at hand points to the more frequent use of the transition zone of caves instead of the dark zone [29,33] [this study]. From all 4170 culicid specimens collected by D. Weber in a period from 2007 to 2015, 69% were collected in the transition zone, 29% in the dark zone, and only a minority of 2% in the entrance zone (Appendix A) [16]. Cave tourism may affect if and how mosquitoes are able to use subterranean habitats. Recent data indicate that higher hibernation mortality may result in this observation. Lipid reserves of overwintering *Culex* females suggest a hibernation temperature optimum ranging from 0 to 8 °C; disturbance (i.e., warming, predator attacks or human activity) interrupts hibernation and lead to increased energy demand, either by increased metabolic rates or flight activity [47], and in turn increases mortality of overwintering females. Anecdotal evidence suggests that mortality in hibernating mosquito females is generally high, and an increase may have deleterious effects on cave mosquito communities [33]. In this context, artificial urban hibernation shelters may be of greater importance for mosquito populations—and the associated vector-borne pathogens—if tourism to natural caves continues to grow.

## 4. Materials and Methods

In Germany, mosquitoes were retrieved and selected from a larger ongoing metabarcoding project investigating the effect of tourism on subterranean invertebrate communities. A total of 12 subterranean sites in Franconian Switzerland (2), Harz (2), Süntel (2), Swabian Jura (4), Westerwald (2) were visited, comprising of paired sets of natural and caves equipped for touristic activities. Over a period of 13.5 months between autumn 2017 and autumn 2018, all sampling sites were visited twice in each season (autumn, spring, summer). At the first visit, specimens were actively collected, and ethanol-filled Barber traps were installed to preferably capture ground-dwelling fauna. This was done separately at an outside reference, in the transition zone and in the dark zone of the cave. At each second seasonal visit, Barber traps were re-collected and long-term Barber traps until the next season were installed. Mosquitoes were sampled by hand using an ethanol-wetted brush, while only a single specimen was collected in a Barber trap (sample ID SubCul006, Appendix A).

For Luxembourg, previously undetermined material from the collection of D. Weber was investigated [16] and further ongoing collection material was included. The material was collected by hand (i.e., using a wetted brush or a vial) between 2007 and 2015. Due to non-optimal storage conditions, DNA quality was often very low, and it was thus only possible to successfully isolate enough DNA for the analyzed specimens.

DNA was extracted from one leg of each mosquito using a modified CTAB-protocol (for Austrian specimens) or one to several legs and a modified salt extraction protocol (for German and Luxembourg specimens). For all three regions, the molecular identification of *Cx. pipiens* forms and *Cx. torrentium* was performed as described in Zittra et al. [2,33], following the protocol by Bahnck and Fonseca [12]: First, *Cx. pipiens* f. *pipiens*, *Cx. pipiens* f. *molestus* and their hybrids were distinguished from *Cx. torrentium* by partial amplification (GoTaq G2 Hot Start Polymerase, Promega GmbH, Walldorf, Germany) of the ACE2 gene using the primers (synthesized at Sigma-Aldrich/Merck KGaA, Darmstadt, Germany) ACEpip, ACEpall, ACEtorr, and B1246s [13]. PCR products were separated using gel electrophoresis targeting 634 bp (*Cx. pipiens* forms) and 512 bp (*Cx. torrentium*) DNA fragments (peqGOLD agarose, VWR International LLC, Vienna, Austria). In the following step, mosquitoes were identified as taxa belonging to the *Cx. pipiens* complex were further identified to form using primers (synthesized at Sigma-Aldrich/Merck KGaA, Darmstadt, Germany) CQ11F2, pip CQ11R, and mol CQ11R. PCR products were visualized using gel electrophoresis targeting 185 bp (*Cx. pipiens* f. *pipiens*) and 241 bp (*Cx. pipiens* f. *molestus*) DNA fragments [12].

Available literature was obtained through specialized searches on GoogleScholar, using combinations of the following keywords: Mosquitoes, Diptera, Culicidae, *Aedes*, *Anopheles*, *Culex*, *Culex pipiens* complex, *Culex pipiens* assemblage (including taxon-specific queries), *Coquillettidia*, *Culiseta*, *Mansonia*, *Ochlerotatus*, *Orthopodomyia*, *Uranotaenia*, caves, subterranean, underground and Europe, and supplemented by grey literature.

## 5. Conclusions

Investigations on the forms of the common house mosquito *Cx. pipiens* in the UK underground channels in 1998 [10] were interpreted as indicating behavioral and ecological differentiation resulting in reproductive isolation of the forms. Our data demonstrates a comparable distribution of *Cx. pipiens* forms in above- and underground habitats. Additionally, caves harbor specific mosquito communities, even though some mosquito species are to be considered subtroglophilous. Among those are primary vectors of mosquito-borne pathogens such as the West Nile virus. Consequently, the significance of subterranean habitats in vector-pathogen dynamics should be fully explored and species unambiguously identified—it is possible that caves are primary reservoirs of vector-borne pathogens, and their epidemiological significance as well as population dynamics of individual mosquito species need to be assessed.

## Figures and Tables

**Table 1 pathogens-10-01090-t001:** The number of individuals and percentage of the *Culex pipiens* complex taxa and *Culex torrentium* sampled in Germany and Luxembourg (this study) in comparison to Austria [33]. Provided are the total number of genetically analyzed specimens per species and their frequencies in the total regional datasets.

Country	*Cx. p.* f. *pipiens*	*Cx. p.* f. *molestus*	*Cx. p*. f. *pipiens* X f. *molestus*	*Cx. torrentium*	Total (*n*)
Germany	99 (66%)	2 (1%)	4 (3%)	46 (30%)	151
Luxembourg	119 (75%)	2 (1%)	0	38 (24%)	159
Austria	44 (34%)	3 (2%)	13 (11%)	69 (53%)	126

**Table 2 pathogens-10-01090-t002:** Mosquito species detected in artificial or natural subterranean shelters.

Taxon	AT *	CZ	DE *	FR	HR	HU	IT	LU *	NO	PL	SE	SK
*Aedes cinereus*/ *geminus*			x									
*Ae. cataphylla*			x									
*Ae. communis*			x									
*Ae. geniculatus*	x											
*Ae*. *rossicus*			x									
*Anopheles* *maculipennis* s. l.	x	x	x	x		x						
*An. messeae*		x				x						
*An. claviger*			x									
*Culiseta* *alaskaensis*		x	x						x	x	x	x
*Cs. annulata*	x	x	x	x		x		x ^1^	x			
*Cs.* *glaphyroptera*		x	x					x ^1^				x
*Culex hortensis*	x			x		x						
*Cx. modestus*	x		x									
*Culex* sp.					x							
*Cx. pipiens* s. l.		x		x			x		x			
*Cx. p.* f. *molestus*	x		x ^1^			x		x ^1^				
*Cx*. *p*. f. *pipiens*	x		x ^1^			x		x ^1^				
*Cx*. *p*. f. *pipiens* X *molestus*	x		x ^1^									
*Cx. territans*	x					x			x			
*Cx. torrentium*	x		x ^1^					x^1^				
*Uranotaenia unguiculata*	x	x				x						

Countries in which species identification was supported by the usage of reliable molecular tools are indicated with *, original data compiled in this study are indicated with ^1^, AT = Austria, CZ = Czechia, DE = Germany, FR = France, HR = Croatia, HU = Hungary, IT = Italy, LU = Luxembourg, NO = Norway, PL = Poland, SE = Sweden, SK = Slovakia.

## Data Availability

All data are available in the supplements to this publication.

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
