# Peer review of "Mosquitoes (Diptera: Culicidae) in the Dark—Highlighting the Importance of Genetically Identifying Mosquito Populations in Subterranean Environments of Central Europe"

_pathogens, 2021, doi:10.3390/pathogens10091090_

Round 1

Reviewer 1 Report

The MS by Zittra et al. aims to describe the distribution patterns of the members of the Cx. pipiens complex in subterranean environments within Central Europe. For that, they provide original data taken in Germany and Luxemburg, and complement it with literature compiled evidence from elsewhere.

Although the work is of interest, the main question is hard to identify and shifts between broad mosquito communities and focusing on the pipiens complex along the MS. This should be better defined and sustained throughout. The Abstract refers solely to the complex (and the mention of Cu. glaphyroptera as first identified in Luxembourg), this is as well the focus of the entire introduction with a few mentions to the Culex genus. However, the aim of the work (lines 102-104) states "to provide a first detailed synopsis on mosquitoes in underground environments with particular focus on the Cx. pipiens complex". The Results section focuses strictly on the Cx. pipiens complex but then you provide Table 2 (in the Discussion section!) which includes the identification of all mosquito species in underground habitats. The information should be reorganized, if you intend to include Table 2 I believe it should be part of the Results section, and the introduction should be broadened to include mosquito communities as a whole.

The information in supplementary tables is valuable but should be summarized somehow in the body of the MS. As presented, it is very difficult to make a picture of if there are differences in the distribution of the members of the complex seasonally, and between natural vs. touristic caves. Regarding the latter, the methodology of collection of specimens in Germany is not clear. From lines 207-208 it appears that you sample paired caves, one of them "pristine" and the other "exploited for tourism". Then in Table S2 the column says "artificial/natural/touristic" but then all the samples are either "natural" or "natural, touristic". Please clarify. Also, if the main conclusion of the work is "our data demonstrate a comparable distribution of Cx. pipiens forms in above- and underground habitats" (lines 240-241), the data supporting this conclusion should be in the body of the MS, not disaggregated in a supplementary table.

Maybe a map of Central Europe would be nice, pointing at the study places herein and the other locations in which previous work has been performed, and some sort of pie chart showing the relative abundance of the members of the complex.

Minor comments

Abstract

lines 22, 26. Abbreviate Cx. after first use.

line 26. "rarely found mosquito taxa". What do you mean? not clear.

Introduction

lines 42-44 vector communities link vectors and hosts? this sounds weird. Vector links hosts-hosts or reservoirs-hosts, or do you refer to the link pathogen-host? please clarify.

line 45 between --> among

lines 74-77 word repetitions. connected / connections, suitable / suitable

line 87 "this practical is not ideal". what do you mean? not clear.

lines 90-91 needs a reference.

line 100. "access vector-borne dynamics..." sounds weird. please rephrase.

Results

italics missing in lines 106, 108, 110, 113, 114 and so on.

line 116. opening parenthesis misplaced

Table 1. Austria. No specifications whatsoever on this work and why it is included here!

Discussion.

The information given at the beginning of this section should be in the Results section!! It is the aim of your study to "review available literature and present new data" (line 102) therefore what you found in the reviewing process is your first result.

lines 148-149. As the information is presented, this statement is not reflected in Table 1.

lines 156-157 "strongly deviating patterns would have been expected in underground habitats" Isn´t that what you found indeed?

line 164. Which assessment? not clear

line 178 reference 36 twice

lines 179-180 "the cave habitats and their characteristics are poorly described". I must say that this statement (that I assume you mention as a flaw of the available literature) can be perfectly applied to the present MS.

line 191 carves --> caves?

M&M

line 219. what do you mean by collection "by hand"? please clarify

line 221 "many specimens" how many?

lines 235-236 can you please give more detail on the search (e.g. keywords, languages, repositories...)

Although the MS is moderately well-written, grammar (specially verb tenses) should be revised (e.g. "is --> has been" line 73, "had been --> has been" line 97, "could be --> were" line 113, "were --> was" line 129).

Author Response

Comment 1: Although the work is of interest, the main question is hard to identify and shifts between broad mosquito communities and focusing on the pipiens complex along the MS. This should be better defined and sustained throughout. The Abstract refers solely to the complex (and the mention of Cu. glaphyroptera as first identified in Luxembourg), this is as well the focus of the entire introduction with a few mentions to the Culex genus. However, the aim of the work (lines 102-104) states "to provide a first detailed synopsis on mosquitoes in underground environments with particular focus on the Cx. pipiens complex". The Results section focuses strictly on the Cx. pipiens complex but then you provide Table 2 (in the Discussion section!) which includes the identification of all mosquito species in underground habitats. The information should be reorganized, if you intend to include Table 2 I believe it should be part of the Results section, and the introduction should be broadened to include mosquito communities as a whole.

Response: We agree with the reviewer and made minor changes to streamline the manuscript. As suggested, Table 2 is now part of the Results, but the scope was rephrased to focus on the Cx. pipiens complex. The new scope reads “. In this contribution, we review available literature and present new data to provide a first synopsis on mosquitoes of the Cx. pipiens complex in underground environments, and provide a summary of Culicidae in subterranean habitats.” The manuscript was modified to reflect this change.

Comment2: The information in supplementary tables is valuable but should be summarized somehow in the body of the MS. As presented, it is very difficult to make a picture of if there are differences in the distribution of the members of the complex seasonally, and between natural vs. touristic caves. Regarding the latter, the methodology of collection of specimens in Germany is not clear. From lines 207-208 it appears that you sample paired caves, one of them "pristine" and the other "exploited for tourism". Then in Table S2 the column says "artificial/natural/touristic" but then all the samples are either "natural" or "natural, touristic". Please clarify. Also, if the main conclusion of the work is "our data demonstrate a comparable distribution of Cx. pipiens forms in above- and underground habitats" (lines 240-241), the data supporting this conclusion should be in the body of the MS, not disaggregated in a supplementary table

Response: We largely removed the references to tourism as this is not the focus of the present study. Thus, we feel there is no need to further present or discuss the details related to in the first part of this comment. Concerning seasonal distribution, there simply is not sufficient data for a proper analysis and our manuscript does not attempt to do so.

Concerning the second part of this comment, relating to distributions of Cx. pipiens forms, we are certain that all relevant information is provided in the manuscript. We present “subterranean” data on this complex from three different countries (Austria, Germany, Luxembourg); there is a suite of studies on relative frequencies of Cx. pipiens forms and their hybrids in Europe (e.g. reference 2 and subsequent studies) reporting similar proportions that can be easily found and reviewed.

Comment3: Maybe a map of Central Europe would be nice, pointing at the study places herein and the other locations in which previous work has been performed, and some sort of pie chart showing the relative abundance of the members of the complex.

Response: Originally, we wanted to include such a file, but the speleological associations and authorities are not in favor of publishing cave or cave entrance coordinates. This is because uncontrolled and sometimes illegal cave explorations have in the past destroyed caves, and led to the extinction of single populations of cave-dwelling species. Thus, we refrain from providing coordinates to these sites, in an attempt to help preserving these sensitive ecosystems.

Minor comments

Abstract

Comment1: lines 22, 26. Abbreviate Cx. after first use.

Response: Changed as suggested.

Comment2: line 26. "rarely found mosquito taxa". What do you mean? not clear.

Response: During the screening of the sampled mosquitoes for the presence of the members of the Culex pipiens complex, we detected some usually rarely found mosquito taxa (that we did not expect to find in caves). It was not described in detail in the abstract, as the number of words is limited. But we now changed the text: “Herein, we review the underground community structures of this species complex in Europe, add new data to Germany and provide first reports of the Culex pipiens complex and usually rarely found mosquito taxa in underground areas of Luxembourg.”

Introduction

Comment1: lines 42-44 vector communities link vectors and hosts? this sounds weird. Vector links hosts-hosts or reservoirs-hosts, or do you refer to the link pathogen-host? please clarify.

Response: Sentence changed: ‘Emergence, distribution and transmission of the West Nile virus and other mosquito-borne pathogens are regulated through potential vector communities that link suitable reservoirs and susceptible hosts.’

Comment2: line 45 between --> among

Response: changed as suggested

Comment3: lines 74-77 word repetitions. connected / connections, suitable / suitable

Response: Thank you, we changed the sentence: ‘Amongst the earliest reports is that of Legendre [17], who observed the larval development of Culex pipiens s. l. in well-connected underground cave systems with strong exchange with above-ground habitats if water and air temperatures were suitable and nutrients available. Subterranean sites provide stable and adequate conditions’

Comment4: line 87 "this practice is not ideal". what do you mean? not clear.

Response: We describe the situation that Cx. pipiens forms + Cx. torrentium are still often summarized as “Cx. pipiens” despite their potentially different distribution patterns and epidemiological relevance means.

Comment5: lines 90-91 needs a reference.

Response: Reference included: Zittra, C., Flechl, E., Kothmayer, M.; Vitecek, S.; Rossiter, H.; Zechmeister, T.; Fuehrer, H.P. Ecological characterization and molecular differentiation of Culex pipiens complex taxa and Culex torrentium in Eastern Austria. Parasite. Vectors. 2016, 9, 197.

Comment6: line 100. "access vector-borne dynamics..." sounds weird. please rephrase.

Response: Thank you for finding this typing error, it was changed in ‘crucial to assess vector-borne pathogen dynamics’

Results

Comment1: italics missing in lines 106, 108, 110, 113, 114 and so on.

Response: Changed throughout the manuscript.

Comment2: line 116. opening parenthesis misplaced

Response: Corrected.

Comment3: Table 1. Austria. No specifications whatsoever on this work and why it is included here!

Response: No specifications because it is cited, already published literature and the only publication where the results can be compared directly (same molecular method). But the information was incorporated ‘Countries where morphological identification and the use of molecular tools were combined (Austria, Germany) but data on numbers of molecularly identified specimen numbers are provided in here (Supplementary Table S1, S2) and the Austrian study [33] only.’

Discussion.

Comment1: The information given at the beginning of this section should be in the Results section!! It is the aim of your study to "review available literature and present new data" (line 102) therefore what you found in the reviewing process is your first result.

Response: We agree with this and moved this part to the results section.

Comment2: lines 148-149. As the information is presented, this statement is not reflected in Table 1.

Response: The information is reflected in the Supplementary Tables, we changed the sentence: ‘We found Cx. pipiens f. pipiens to be the dominating mosquito in the investigated natural and artificial subterranean sites (Supplementary Table S1).

Comment3: lines 156-157 "strongly deviating patterns would have been expected in underground habitats" Isn´t that what you found indeed?

Response: No. We found proportions of Cx. pipiens f. pipiens, Cx. pipiens f. molestus and hybrids to reflect that of previously published above-ground populations. We rephrased the section to make this clearer.

Comment4: line 164. Which assessment? not clear

Response: Changed to: ‘More frequent and more specific sampling in underground habitats, including the collection of potentially present mosquito larvae, should be conducted to assess potential reproductive isolation between Cx. pipiens forms’

Comment5: line 178 reference 36 twice

Response: Removed

Comment5: lines 179-180 "the cave habitats and their characteristics are poorly described". I must say that this statement (that I assume you mention as a flaw of the available literature) can be perfectly applied to the present MS.

Response: This comment is correct (although we do not consider this a flaw of available literature – who can know what parameters to consider?), but there is, alas, no solution to the problem. This work is the first to provide a summary on mosquitoes in caves, which we consider as necessary incentive to conduct more focused ecological studies.

Comment6: line 191 carves --> caves?

Response: Corrected.

M&M

Comment1: line 219. what do you mean by collection "by hand"? please clarify

Response: collection ‘by hand’ means that the specimens have been collected actively with hand, using either tubes or a wetted brush. Culicidae only very rarely can be encountered from passive collection events.

Comment2: line 221 "many specimens" how many?

Response: We have reformulated the sentence now a bit: “Due to non-optimal storage conditions, DNA quality was often very low, and it was thus only possible to successfully isolate enough DNA for the analysed specimens.”

Comment3: lines 235-236 can you please give more detail on the search (e.g. keywords, languages, repositories...)

Response: Done as suggested by the reviewer: ‘Available literature was obtained through specialized searches on GoogleScholar, using combinations of the following keywords: Mosquitoes, Diptera, Culicidae, Aedes, Anopheles, Culex, Culex pipiens complex, Culex pipiens assemblage including taxa specific queries, Coquillettidia, Culiseta, Mansonia, Ochlerotatus, Orthopodomyia, Uranotaenia, Caves, subterranean, underground and Europe, and supplemented by grey literature.’

Comment4: Although the MS is moderately well-written, grammar (specially verb tenses) should be revised (e.g. "is --> has been" line 73, "had been --> has been" line 97, "could be --> were" line 113, "were --> was" line 129).

Response: In addition to the above comments, all spelling and grammatical errors pointed out by the reviewers have been corrected. The manuscript had been checked by a native speaker before resubmission.

Reviewer 2 Report

  • Latin names of mosquitoes in chapter Results (1st and 2nd paragreph) should be in italics.
  • Last para in Dicussion- caves insted of carves

Author Response

All minor points raised by Reviewer 2 were addressed and incorporated in the manuscript.

Round 2

Reviewer 1 Report

Abstract.

Line 16. Hardly or non-distinguishable? I believe that both are correct but you should state your position. Do you trust male genitalia? or female wing morphometrics?

Lines 23 and 26 (again). Abbreviate Culex.

Introduction.

Lines 36-37-I would remove "that is generally anthropophilous", because it cuts the link between the mention to the nominal species and its bioforms, and also because the feeding behaviour of other members of the complex is not described here (first sentence of the work). On top, the preference for birds or mammals depends on the bioform (as properly addressed in lines 53-56).

Line 55. Cx. f. pipiens --> Cx. pipiens f. pipiens

Lines 60-61. Needs a reference

Line 60. Do you refer to the members of the Cx. pipiens complex? or to species from the genus Culex in general? Not clear. (According to the rest of the paragraph it seems the first option).

Line 74. Abbreviate Culex.

Lines 85-86. What does "data entering" mean?

Line 90. In --> Within; seems to be --> is

Discussion

Lines 216-217. Word repetition "cave"

Methods

Line 241. Reference 11 is Bahnck and Fonseca, not Smith and Fonseca.

Authors responses to comments of the former version

Major comments

Comment 1. OK!

Comment 2 holds what I stated earlier.

Comment 3. OK, I understand, although at the suggested spatial scale geographical coordinates would not need to be accurately provided.

Minor comments

Comment 4 holds what I stated earlier. Still unclear in the text.

Author Response

Dear Reviewer 1,

thank you very much for your second round of valuable comments and corrections. Please find below a point by point response to those:

Abstract.

Line 16. Hardly or non-distinguishable? I believe that both are correct but you should state your position. Do you trust male genitalia? or female wing morphometrics?

Both is correct. Some species can be hardly morphologically distinguished (e.g. on male genitalia), others such as Cx. pipiens f. molestus and pipiens (depending on the specimens at hand), are non-distinguishable.

Lines 23 and 26 (again). Abbreviate Culex.

Done.

Introduction.

Lines 36-37-I would remove "that is generally anthropophilous", because it cuts the link between the mention to the nominal species and its bioforms, and also because the feeding behaviour of other members of the complex is not described here (first sentence of the work). On top, the preference for birds or mammals depends on the bioform (as properly addressed in lines 53-56).

We removed the part of the sentence accordingly.

Line 55. Cx. f. pipiens --> Cx. pipiens f. pipiens

Done.

Lines 60-61. Needs a reference

Legendre (1931) added as reference. Reference numbering adjusted accordingly.

Line 60. Do you refer to the members of the Cx. pipiens complex? or to species from the genus Culex in general? Not clear. (According to the rest of the paragraph it seems the first option).

We refer to the member of the Culex pipiens complex. We re-formulated that part.

Line 74. Abbreviate Culex.

Done.

Lines 85-86. What does "data entering" mean?

The data which were the basis for the Dörge et al. (2019) publication and analyses are based on long-term collection records of Culicidae from Hesse. We now re-formulated our statement to make this more clear.

Line 90. In --> Within; seems to be --> is

Sentence corrected.

Discussion

Lines 216-217. Word repetition "cave"

removed.

Methods

Line 241. Reference 11 is Bahnck and Fonseca, not Smith and Fonseca.

corrected.

Authors responses to comments of the former version

Major Comment 2 holds what I stated earlier.

We can only repeat our statement from our last revision here, as we see the situation solved in the current version of the manuscript:

“We largely removed the references to tourism as this is not the focus of the present study. Thus, we feel there is no need to further present or discuss the details related to in the first part of this comment. Concerning seasonal distribution, there simply is not sufficient data for a proper analysis and our manuscript does not attempt to do so.

Concerning the second part of this comment, relating to distributions of Cx. pipiens forms, we are certain that all relevant information is provided in the manuscript. We present “subterranean” data on this complex from three different countries (Austria, Germany, Luxembourg); there is a suite of studies on relative frequencies of Cx. pipiens forms and their hybrids in Europe (e.g. reference 2 and subsequent studies) reporting similar proportions that can be easily found and reviewed.”

Minor Comment 4 holds what I stated earlier. Still unclear in the text.

We tried to make our explanation more clear now: "[] the general practice of reporting combined occurrence data for Cx. pipiens s.l. and Cx. torrentium as “Culicidae” or even “Culex pipiens” is not ideal given their potentially different distribution patterns and epidemiological relevance."